# The Effectiveness of ddPCR for Detection of Point Mutations in Poor-Quality Saliva Samples

**DOI:** 10.3390/healthcare10050947

**Published:** 2022-05-20

**Authors:** Petra Riedlova, Dagmar Kramna, Silvie Ostrizkova, Hana Tomaskova, Vitezslav Jirik

**Affiliations:** 1Centre for Epidemiological Research, Faculty of Medicine, University of Ostrava, 70103 Ostrava, Czech Republic; dagmar.kramna@osu.cz (D.K.); silvie.ostrizkova@osu.cz (S.O.); hana.tomaskova@osu.cz (H.T.); vitezslav.jirik@osu.cz (V.J.); 2Department of Epidemiology and Public Health, Faculty of Medicine, University of Ostrava, 70103 Ostrava, Czech Republic

**Keywords:** ddPCR, saliva, blood, manual isolation, point mutation, fractional abundance

## Abstract

Background: The noninvasive collection of saliva samples for DNA analyses is simple, and its potential for research and diagnostic purposes is great. However, DNA isolates from such samples are often of inferior quality to those from blood. Aim: The aim of this study was to investigate the robustness and sensitivity of the ddPCR instrument for genetic analyses from saliva samples of poor quality by comparing their results to those obtained using an established method from blood samples. Methods: Blood and saliva were collected from 47 university students, which was followed by manual isolation of DNA and analysis on droplet digital PCR (ddPCR). Results of analyses were supplemented with values of fractional abundances. Results: ddPCR proved to be highly suitable for analysis of even low-quality saliva samples (concentrations as low as 0.79 ng/µL), especially when augmented by fractional abundance data. This combination yielded 100% agreement with results obtained from blood samples. Conclusion: This study verified the applicability of ddPCR as a sensitive and robust method of genetic diagnostic testing even from low-quality saliva isolates. This makes it potentially suitable for a wide range of applications and facilitates the performance of large epidemiological studies, even if sampling or sample processing is suboptimal.

## 1. Introduction

A wide range of biological samples collected using invasive or noninvasive methods can be used for DNA analysis. Although noninvasive methods are obviously preferable by the subjects [1], invasive methods (especially blood collection) are often considered necessary for the acquisition of a sufficient amount and quality of genomic DNA [2,3]. In practice, however, there are circumstances where blood collection is not possible due to ethical reasons or the subject’s refusal [2,3,4]. Moreover, in epidemiological studies aiming to collect a large number of samples from various locations, the use of a noninvasive biological material can be also beneficial, as it may not need the presence of a healthcare worker [2,4,5]. Saliva samples are a good candidate for such noninvasive DNA sample collection for research and diagnostic purposes [6], not least thanks to the availability of cheap, easy-to-use, and commercially available sampling sets [7]. An additional benefit of saliva compared to blood is the lower risk of transfer of infection to laboratory workers [8]. Various collection methods, biological materials, and analyte (including DNA) isolation approaches have been evaluated in the literature, e.g., [2,3,5,7,9], but the choice of the particular approach largely depends on the endpoint measurement, i.e., on the amount of the isolated analyte necessary for detection and quantification by the particular analytical method. Obviously, given the generally lower DNA concentration in saliva, the quality of DNA isolation is another key factor affecting the method performance. In saliva, a significantly smaller amount of DNA and a sample of generally lower quality than in blood can be expected; in effect, the DNA isolates from saliva may be below the level of detection for some classical methods of DNA analysis [5]. For this reason, the suitability of a highly accurate and sensitive endpoint method capable of absolute quantification, namely droplet digital PCR (ddPCR) [10,11], for the analysis of saliva samples was evaluated in this study. This work was linked to a study on the risk of thrombophilic mutations in young women [12] performed within the scope of a multifaceted project HAIE (Healthy Aging in Industrial Environment); the confirmation of the suitability of this method could lead to a greater comfort of patients during sampling, and thus help in screening and/or recruiting patients for larger epidemiological studies.

This study aimed to find out the robustness and sensitivity of the ddPCR method for detection of the *Factor V Leiden* mutation in saliva samples manually isolated by university students, which simulates highly adverse circumstances for any genetic analysis by comparing the results of this method to those of the analysis performed from the blood samples from the same subjects using a previously validated method.

## 2. Materials and Methods

### 2.1. Sample Collection

Saliva samples were collected from 47 female university students aged 18–37 years using a special pad that was inserted into the mouth for 3 min, and participants were asked to move it around in their mouth; subsequently, the pad was placed into special tubes that were centrifuged (1000 g/2 min/20 °C), and the supernatant was transferred to a 1.5 mL tube and frozen at −24 °C until DNA isolation. At the same time, venous blood was collected from each student into 2.5 mL tubes with EDTA, left at room temperature for at least 2 h, and subsequently, frozen at −24 °C until the time of DNA isolation.

### 2.2. Statistical Analysis of Isolated Samples

DNA isolation from saliva samples was performed manually using the CatchGene Saliva DNA kit (CatchGene, New Taipei City, Taiwan) within the scope of practical training of students at our faculty. DNA isolation from blood was also performed manually using the NucleoSpin Blood kit (Macherey-Nagel, Dueren, Germany) as described in our previous paper [12]. Quality control of isolated DNA was performed by measuring the absorbance ratio of the extract at 260 to 280 nm (A260/280) on a microvolume spectrophotometer (DeNovix, Wilimington, DE, USA), and the concentration of isolated DNA in ng/uL was established spectrophotometrically. High variability in both the concentration and purity of the isolated samples was expected due to the isolation by students learning the procedure; should ddPCR from saliva samples be successful despite such adverse circumstances, it would be an important indication of the robustness of the method.

### 2.3. Statistical Sample Analysis

Data were presented using parametric (mean, standard deviation where data were distributed normally) and nonparametric (median, 95% confidence intervals; all data) descriptors. Correlation between concentrations in saliva and blood samples was evaluated using Spearman’s correlation coefficient and the normality of the distribution using Skewness/Kurtosis tests for normality.

### 2.4. Laboratory Sample Analysis

Both the blood and saliva samples were analysed for the presence of a gene encoding the *Factor V Leiden* (*FV*) c.1691G > A mutation [12]. Based on the concentration of the DNA isolate, the amount of the sample and of other reaction-mixture components (DNA template, PCR water, ddPCR supermix, and ddPCR mutassay; Biorad, Hercules, CA, USA) were determined [11]. Two fluorescent-dye-containing hydrolysis probes, FAM (Fluorescein amidite) and HEX (Hexachloro-fluorescein) binding to the target amplicon (rs6025) were used for analysis. The FAM probe is complementary to the mutant allele (MUT), the HEX probe to the wild-type allele (WT). The droplet emulsion was thermally cycled in the following conditions: denaturing at 95 °C for 10 min, 40 cycles of PCR at 94 °C for 30 s and 57 °C for 2 min, and a final extension at 98 °C for 10 min. PCR amplification in droplets was confirmed using the QX200 Droplet Reader (Bio–Rad) [11].

Fluorescence measurement of the final PCR product, i.e., up to 20,000 droplets containing MUT and WT templates, was performed using the number of positive (fluorescent) droplets and their fluorescence amplitude; after software processing, including Poisson statistics, they were expressed as the numbers of copies of the templates (of the selected DNA segment) with point mutation and those of “wild-type” templates (copies/sample). In addition, the copy numbers of both templates per DNA amount in the PCR mixture (copies/ng DNA), respectively, were also calculated.

### 2.5. Statistical Sample Analysis

Test parameters (false-negative rate, false-positive rate, sensitivity, specificity, positive and negative predictive value, accuracy) were calculated in the usual way [13].

To improve the sensitivity and for confirmation of the positivity/negativity of the results, an additional analysis using fractional abundance was performed. Fractional abundance (FA%) was calculated from the numbers of copies of MUT and WT according to Equation (1) [14]:FA = 100 × C_MUT_/(C_MUT_ + C_WT_) [%](1)
where C_MUT_ is the number of copies of the DNA template with a point mutation per 1 ng DNA of the sample (20 µL) and C_WT_ is the number of copies of the DNA template without point mutation (wild type) per 1 ng DNA of the sample (20 µL).

In heterozygotes, i.e., where both MUT and WT templates are present in the sample, the expected FA is approx. 50%; FA values in samples from homozygous individuals should converge to 100% and values from samples containing only WT alleles should be about 0%.

## 3. Results

The purity and concentration of DNA in the student-prepared extracts from both types of biological samples are shown in Table 1.

As expected, no correlation was found between DNA isolates from blood and saliva.

Figure 1 presents results of the analyses of (a) blood and (c) saliva samples using ddPCR, as well as results of fractional abundance calculated from (b) blood and (d) saliva samples.

Parameters showing the performance of the ddPCR-based analysis of saliva isolates for prediction of blood-based results (considered true values) evaluated by a combined method of the determination of the number of copies and fractional abundance are presented in Table 2.

## 4. Discussion

This study aimed to find out the robustness and sensitivity of the ddPCR method in a population of young women (young women were selected in view of the fact that the use of hormonal contraceptives in women with Factor V Leiden mutation significantly increases the risk of thromboembolic events) [12]. To account for the worst possible scenario, we used saliva samples (with generally lower DNA content) manually processed by students of the Faculty of Medicine; the quality of such samples was expected to be poor, which was confirmed by the low DNA concentration and purity. Results of the previously established method using blood samples were used as a reference.

The ddPCR method proved to be suitable even for such poor DNA. This opens possibilities for the use of this method for a wide range of epidemiological studies with noninvasive sampling not requiring the presence of healthcare personnel during sample collection.

Both the concentration and purity of saliva-based DNA isolates in our study were highly variable; this variability was lower for both parameters where blood-based isolates were concerned (see Table 1). The higher variability of DNA concentration in saliva samples can be caused by differences in DNA concentration in the saliva samples (differences in saliva secretion and sampling) as well as by manual isolation by university students; automated isolation would likely yield better and less variable results [9]. In addition, saliva (unlike blood) can also contain substances potentially interfering with the analysis [15]. The purity of blood samples in our study was 1.78, thus being close to the ideal value of 1.8 [15]. In saliva, the mean (1.67) and especially the median (1.5) purity was—as expected—far from ideal, indicating major protein contamination of DNA extracts (Table 1). Differences in purity between saliva and blood have been reported in other studies as well. Hansen et al. [2] and Abraham et al. [15] reported means of 1.71–1.79 from blood and 1.56–1.63 from saliva samples, respectively, which was similar to our results. On the other hand, Looi et al. reported comparable purity of both sample types, with 1.69 in blood and 1.71 in saliva [9]. Although our study participants were informed to avoid using chewing gum, eating, drinking, or smoking, adherence to this advice is not 100% assured. Similar to the study by Williamson et al. [5] or Dillon et al. [16], no significant correlation between DNA amount obtained from blood and saliva samples was observed (Table 1).

The acquired DNA concentrations were in some cases suboptimal, preventing analysis by classical PCR. This supports the necessity of the use of an endpoint analysis with maximum sensitivity, which the ddPCR method proved to be. Comparison of the performance of ddPCR to that of other molecular biological methods is highly favourable. For example, Hansen et al. were able to use all 32 blood samples for analysis using standard PCR combined with Sanger sequencing, while only 84% out of 72 saliva samples were suitable for PCR analysis [2]. In another study, the limits of detection (LoD) for real-time PCR (qPCR), next-generation sequencing (NGS) and ddPCR was compared. The detection limit for ddPCR was only 0.01% compared with 0.12% for qPCR and as much as 2–6% for NGS [17,18]. In our study, thanks to the use of ddPCR, we were able to use 100% of blood samples as well as of the isolates from saliva samples, despite the suboptimal DNA isolation.

The results of the analysis were evaluated as the number of positive droplets and their fluorescence, which was recalculated and expressed as template copy numbers per ng DNA in the isolated sample (Figure 1). The analysis using these parameters alone returned one false-negative result in saliva samples (compared to the blood results). This reduced the accuracy of saliva results to 46 out of 47, i.e., 97.9% (95% IS: 88.7–99.9%) (Table 2). The reason for this false-negative result was probably due to the very low DNA concentration of only 0.79 ng/µL of extract; coincidentally, it was the sample with the lowest DNA concentration in the entire set. Although the declared minimum amount of DNA suitable for ddPCR analysis on the used instrument is 0.6 ng of DNA/µL of extract [11], it is possible that this low concentration could still have been the source of the false negativity.

This false-negative result turned our attention to an additional parameter that can be helpful when analysing saliva samples with a very low DNA content (<1 ng/uL)—fractional abundance. The additional use of this parameter proved to be able to detect this “hidden” false negative, and therefore, it can be considered as a salvage method in the case of poor-quality sample isolation. As can be seen from Figure 1a,c we can see a discrepancy in one sample where a mutant allele has been found in the blood samples, but this was not found in saliva. When evaluated using only the detected copy number, this result therefore did not correspond between blood and saliva. However, FA calculation in this sample revealed a value of 51.8% in blood and 40.0% in saliva; based on the latter result, the sample was declared positive as FA of all true-negative samples was below 5%. Hence, we can confirm that the use of FA can lead to revealing the false-negative results obtained by the classical method. Using fractional abundance, we achieved a 100% agreement of results in our ddPCR method in blood and saliva samples, which indicates not only the good accuracy of the method but also its high robustness. Our results are apparently in agreement with those of other laboratories; for example, Abraham et al. also found a >99% accuracy in blood and >97% in saliva samples RT-PCR [15].

FA poses an interesting option for further improvement of the ddPCR detection limit in heterozygotes. While for a positive signal, a certain number of positive droplets (i.e., droplets with mutated allels) must be present (the exact number is determined in each run), FA allows us to use an even lower number of positive droplets. If the number of droplets with wild-type allele is approximately the same as of those with mutated allele, we know with high reliability that the individual is a heterozygote. To illustrate it on an example, if 30 “mutated allele” droplets must be detected in a sample to be able to determine its positivity in the usual way but only 10 are detected in a sample with low DNA concentration, we can look at the number of droplets with wild-type allele. If the number of those droplets greatly differs from 10, we can say that the 10 droplets represent just random noise. If, however, there are 9 to 11 droplets with WT allele, we can see that the number of mutated and wild-type alleles is about the same, which corresponds to a heterozygote (in other words, we basically use both strands of DNA for confirmation, thus improving the sensitivity) [11].

In view of the costs of ddPCR analysis, which is comparable to other PCR methods and its sensitivity (note that after including FA, sensitivity/specificity, positive and negative predictive values were all at 100%), we can definitely recommend this method in combination with saliva sampling, although obviously, a professional DNA isolation by experienced personnel or automatised isolation are definitely preferable. In addition, other methods of saliva-sample collection could be worth considering; besides the pad used for collection in our study, spitting containers [2,15] or other types of pads [3] could also improve the total amount of DNA in the sample.

In epidemiological studies, the willingness of subjects to participate is crucial for the acquisition of valid data. A pilot study by Hansen et al. investigated how the collection method affects the volunteers’ response rate and willingness to participate in the study; about 75% of volunteers submitted a sample from a noninvasive collection, while when an invasive collection was necessary, less than 33% responded [2]. This was confirmed by results from other studies as well [19].

When deciding on the type of sampling material, it is important to consider analyses for which the samples will be further used. In our case, saliva proved useful for genetic analyses. The collection of blood samples represents more or less certainty of trouble-free analysis, with a generally better DNA purity, and in most cases, a higher concentration of isolates. On the other hand, it is an invasive collection that requires the presence of trained medical personnel. The collection, transport, and storage of blood samples is more complex than that of saliva [2,9,11]. Several studies have confirmed the high DNA quality of saliva samples even after 30 days of storage at 37°C and up to 8 months at room temperature [20]. Therefore, assuming the use of a ddPCR instrument, saliva appears to be a highly suitable alternative to blood sampling for genetic analyses [2,9,11]. Since the collection itself, as mentioned above, is easier, ethical considerations are more straightforward, and the cost of sample testing is similar, saliva (especially in combination with a highly sensitive method such as ddPCR) appears to be suitable for use in large epidemiological studies [2].

An obvious limitation of the presented study lies in the low number of samples, leading to the wide confidence intervals in the evaluated test parameters.

## 5. Conclusions

In this study, we demonstrated the good performance of ddPCR for analysis of (even poorly isolated) saliva samples, proving this method to be highly sensitive and robust for determining the presence of a point mutation. The method shows good potential to be used in epidemiological research, where the use of noninvasive sample collection methods—in this case, saliva—may significantly simplify the recruitment of volunteers, as the noninvasive collection of biological material is generally much more comfortable for participants. The combination of such sampling with the absolute quantification using the highly sensitive ddPCR method, especially in combination with the calculation of fractional abundance, represents a potentially very powerful tool for conducting large-scale epidemiological studies.

## Figures and Tables

**Figure 1 healthcare-10-00947-f001:**
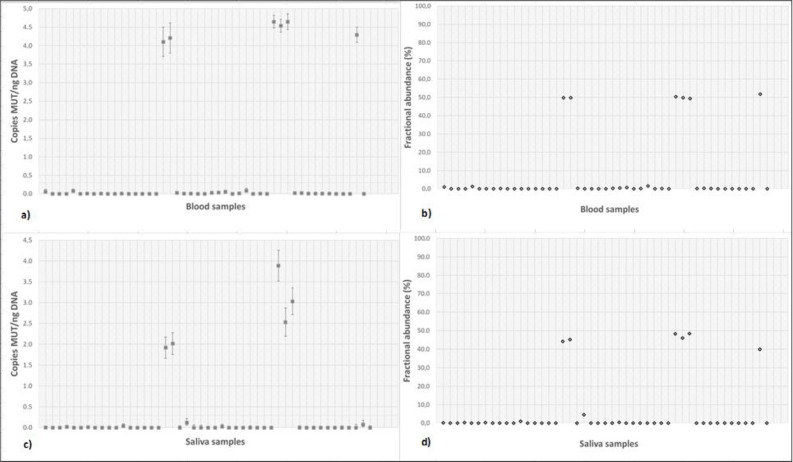
Numbers of copies of the mutated allele detected in blood (**a**) and saliva (**c**) samples using ddPCR, as well as fractional abundance calculated from blood (**b**) and saliva (**d**) samples.

**Table 1 healthcare-10-00947-t001:** Purity and concentration of isolated DNA by students from saliva and blood samples.

	Blood—Extract Purity (A260/280)	Blood—DNA—Concentration (ng/µL)	Saliva—Extract Purity (A260/280)	Saliva—DNA—Concentration (ng/µL)
Number of samples	47	47	47	47
Median (IQR)	1.80 (1.7–1.84)	29.20 (17.6–43.7)	1.50 (1.26–2.13)	5.13 (3.06–10.99)
Arithmetic mean ± SD	1.78 ± 0.11	32.00 ± 17.80	1.67 ± 0.59	N/A
Minimum	1.52	8.50	0.76	0.79
Maximum	2.02	86.00	3.09	54.27
Skewness/Kurtosis tests for Normality	0.7805	0.0629	0.1996	<0.001

IQR—Interquartile range; SD—standard deviation.

**Table 2 healthcare-10-00947-t002:** Basic test parameters for saliva-based ddPCR detection of *FV* mutation using both the simple number of copies (left) and fractional abundance (right); 95% confidence intervals are shown in the brackets. Please note that results from blood samples were used as reference (100%) and absolute numbers were identical to those described for evaluation using fractional abundance.

	Evaluation Using the Number of Copies		Evaluation Using Fractional Abundance	
	Positive (Mutation Present)	Negative (without Mutation)	Positive (Mutation Present)	Negative (without Mutation)
Positive (mutation present)	a = 5	b = 0	a = 6	b = 0
Negative (without mutation)	c = 1	d = 41	c = 0	d = 41
Sensitivity	83.3% (35.9–99.6)		100% (54.1–100)	
Specificity	100% (92.0–100)		100% (91.4–100)	
Positive predictive value	100% (47.8–100)		100% (54.1–100)	
Negative predictive value	97.6% (87.7–99.9)		100% (91.4–100)	
False positive rate	0%		0%	
False neagtive rate	21%		21%	
Accuracy	97.9% (88.7–99.9)		100% (92.4–100)	

## Data Availability

The data presented in this study are available on request from the corresponding author. The data are not publicly available due to ethical and privacy restrictions.

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
