# Peer review of "The Effectiveness of ddPCR for Detection of Point Mutations in Poor-Quality Saliva Samples"

_healthcare, 2022, doi:10.3390/healthcare10050947_

Round 1
Reviewer 1 Report
Author demonstrated the effectiveness of ddPCR for detection of point mutations in poor quality saliva samples as compared to the blood isolated DNA samples.
This is newer technique need to be validated with adequate sample size.
Author should come with the threshold level of DNA for analysis of mutations instead of comparing the poor quality of DNA isolated from saliva.
It is better is the author compare the endpoint results with different methods of DNA isolation instead of stating poor quality as poor quality is a unscientific word.
Author Response
Response to Reviewer 1 Comments
Point 1:
Author should come with the threshold level of DNA for analysis of mutations instead of comparing the poor quality of DNA isolated from saliva.
Response 1: Manufacturer-reported threshold level for our dPCR machine is 0.6 ng/ul isolated DNA. In our study, the DNA concentration in the least concentrated sample was 0.79 ng/ul and we were able to detect it correctly but only after the use of fractional abundance.
However, we do not want to make any strong conclusions about the limit of detection – there were far too few positive samples for reliable modelling of the sigmoidal curve and determination of the limit of detection.
Point 2:
It is better is the author compare the endpoint results with different methods of DNA isolation instead of stating poor quality as poor quality is a unscientific word.
Response 2: This is an excellent point and we absolutely agree. We also intend to compare the manual isolation to the automatic one – however, this is a subject of the next project, this one was a pilot study aiming at determining the usability of samples processed by inexperienced personnel (students).
We apreciate that "poor" is not a scientifically proper word but in this case, we believe that it is justified as we cannot find any word that would not be vague and at the same time describe the whole range of concentrations (we cannot say "low concentration samples" as many had good enough DNA sufficient concentration). Had we omitted the "poor" from the title, it would not emphasize the fact that we focused on samples that were suboptimally processed. The last mentioned term ("suboptimally processed") could replace the word "poor" in the title but still would be vague.
Reviewer 2 Report
- Please indicate the sample representativeness using saliva samples collected from 47 female university students aged 18-37 years in the section of “Material and Method”. Is there any special consideration in gender and age for the samples?
- Please provide the limit of detection (LOD) and limit of quantitation (LOQ) of the ddPCR-based analysis of saliva isolates in the section of “Result”.
- Please indicate how to evaluate the percentages of false positive and false negative cases in the sections of “Material and Method”, and show the results in the section of “Result”.
- In the section of “Discussion”:
(1) Please discuss how to maintain the balance between sensitivity and accuracy (make both sensitive and accurate).
(2) Please compare the the significance, advantages and difference using ddPCR to detect of point mutations in poor quality saliva samples with other methods such as real-time PCR and next generation sequencing (NGS).
Author Response
Response to Reviewer 2 Comments
Point 1:
Please indicate the sample representativeness using saliva samples collected from 47 female university students aged 18-37 years in the section of “Material and Method”. Is there any special consideration in gender and age for the samples?
Response 1: The knowledge of Factor V Leiden mutation is especially important in association with hormonal contraception as its use in women with Factor V Leiden significantly increases the risk of thromboembolic events. For this reason, we addressed only young women (all were university students). We have added this information into the Discussion.
Point 2:
Please provide the limit of detection (LOD) and limit of quantitation (LOQ) of the ddPCR-based analysis of saliva isolates in the section of “Result”.
Response 2: The manufacturer-reported threshold level for our dPCR machine is 0.6 ng/ul isolated DNA. In our study, the DNA concentration in the least concentrated sample was 0.79 ng/ul and we were able to detect it correctly but only after the use of fractional abundance.
However, we do not want to make any strong conclusions about the limit of detection – there were far too few positive samples for reliable modelling of the sigmoidal curve and determination of the limit of detection. If you insist, we can attempt to calculate LOD/LOQ but the results will be indeed unreliable and would, in our opinion, rather decrease the quality of the paper by stating hard numbers that would be unjustified. The information that we were able to detect a sample with 0.79 ng/ul, which roughly corresponds to the manufacturer-declared value, is, in our opinion, more honest in this case.
Point 3:
Please indicate how to evaluate the percentages of false positive and false negative cases in the sections of “Material and Method”, and show the results in the section of “Results”.
Response 3: We added the false positive and false negative rates into the Methods and into the Table 2 in the Results. We have also added a citation where the calculation of these parameters is described (we believe that it would be excessive to state detailed calculation of such basic statistical values in the paper).
Point 4:
In the section of “Discussion”:
(1) Please discuss how to maintain the balance between sensitivity and accuracy (make both sensitive and accurate).
Response 4: With accuracy and sensitivity of 100 % combined with false negative/positive rates of 0 % in our study, even on saliva samples that were manually processed by inexperienced students, the balance cannot be much better, although a better isolation would definitely help. We added the following into the Discussion:
In view of the costs of ddPCR analysis, which is comparable to other PCR methods and its sensitivity (note that after including FA, sensitivity/specificity, positive and negative predictive values were all at 100 %), we can definitely recommend this method in combination with saliva sampling, although, obviously, a professional DNA isolation by experienced personnel or automatized isolation are definitely preferable.
Point 5:
(2) Please compare the the significance, advantages and difference using ddPCR to detect of point mutations in poor quality saliva samples with other methods such as real-time PCR and next generation sequencing (NGS).
Response 5: You are right, the comparison was indeed insufficient. We have added the following into the Discussion:
……Comparison of the performance of ddPCR to that of other molecular biological methods is highly favour-able. For example, Hansen et al. were able to use all 32 blood samples for analysis using standard PCR combined with Sanger sequencing while only 84 % out of 72 saliva sam-ples were suitable for PCR analysis. In other studies, were compared the limit of detection (LoD) for Real Time PCR (qPCR), Next generation sequencing (NGS) and ddPCR. For qPCR the detection limit was 0.12%, for NGS 2-6% and for ddPCR only 0.01%. In our study, thanks to the use of ddPCR, we were able to use 100% of blood samples as well as of the isolates from saliva samples, despite the suboptimal DNA isolation.
Reviewer 3 Report
This manuscript by Petra and colleagues elegantly showed the use of a non-invasive method to collect biological material and isolate DNA from saliva compared to blood. The isolation of DNA from saliva can also be a beneficial incentive that can help facilitate the screening and/or recruitment of subjects for epidemiological research. In this study, the authors measured the quantity and quality of isolated DNA. They used a reliable and sensitive PCR method, namely the droplet digital PCR (ddPCR), to detect the presence of Factor V Leiden mutation in saliva samples compared to the gold standard blood isolation.
DNA isolation is a determining factor in large epidemiological studies; the authors used a suboptimal method to isolate salivary DNA by untrained university students to test its limitations. Both the concentration and purity of saliva-based DNA isolates were highly variable compared to the standard method of blood isolation. Luckily, the ddPCR results further proved that saliva can reliably replace blood for genomic analysis. However, it is appropriate to raise a few questions:
- discuss the discrepancy of the number of copies per ng DNA and fractional abundance of the mutant allele calculated
- Although the data presented in Table 2 are strong and encouraging, it is appropriate to have a similar analysis using blood samples for comparison
- The presence of a false negative data point in the saliva sample with the lowest DNA concentration showed the limitations of DNA sampling from saliva. What are the alternatives to homogenize the process and reduce the large variability?
Author Response
Response to Reviewer 3 Comments
Point 1:
Discuss the discrepancy of the number of copies per ng DNA and fractional abundance of the mutant allele calculated
Response 1:
We have added an explanation for this improvement in sensitivity, attempting to give a clear and simple illustration of the reasons understandable to scientists unfamiliar with ddPCR:
…FA poses an interesting option for further improvement of ddPCR detection limit in heterozygotes. While for a positive signal, a certain number of positive droplets (i.e., droplets with mutated allels) must be present (the exact number is determined in each run), FA allows us to use an even lower number of positive droplets. If the number of droplets with wild-type allele is approximately the same as of those with mutated allele, we know with high reliability that the individual is a heterozygote. To illustrate it on an example – if 30 "mutated allele" droplets must be detected in a sample to be able to determine its positivity in the usual way but only 10 are detected in a sample with low DNA concentration, we can look at the number of droplets with wild type allele. If the number of those droplets greatly differs from 10, we can say that the 10 droplets represent just random noise. If, however, there are 9 to 11 droplets with WT allele, we can see that the number of mutated and wild type alleles is about the same, which corresponds to a heterozygote (in other words, we basically use both strands of DNA for confirmation, thus improving the sensitivity).
Point 2:
Although the data presented in Table 2 are strong and encouraging, it is appropriate to have a similar analysis using blood samples for comparison
Response 2: We have added the following into the description of Table 2:
Please note that results from blood samples were used as reference (100%) and absolute numbers were identical to those described for evaluation using fractional abundance.
Point 3:
The presence of a false negative data point in the saliva sample with the lowest DNA concentration showed the limitations of DNA sampling from saliva. What are the alternatives to homogenize the process and reduce the large variability?
Response 3: This is a fair point. We have added the following into the text:
...a professional DNA isolation by experienced personnel or automatized isolation are definitely preferable. Also, other methods of saliva samples collection could be worth considering – besides the pad used for collection in our study, spitting containers or other types of pads could also improve the total amount of DNA in the sample.
Reviewer 4 Report
The manuscript entitled "The effectiveness of ddPCR for detection of point mutations in poor quality saliva samples" highlighted that the applicability of ddPCR as a sensitive and robust genetic diagnostic testing even from low-quality saliva isolates. This makes it potentially suitable a wide range of applications and facilitates the performance of large epidemiological studies even if sampling or sample processing is suboptimal
- The AUthors should provide the expand forms for all acronyms, including gene acronysm ,through the text when they first appear.
- Gene acronyms should be written in italics.
Author Response
Response to Reviewer 4 Comments
Recenzent 4
Point 1:
The Authors should provide the expand forms for all acronyms, including gene acronysm, through the text when they first appear.
Response 1: We added the last missing acronyms (FAM and HEX) into the text.
Point 2:
Gene acronyms should be written in italics.
Response 2: Thank you, we amended the text accordingly.
Round 2
Reviewer 2 Report
The manuscript has been significantly improved.